# Registered report: Coadministration of a tumor-penetrating peptide enhances the efficacy of cancer drugs

Irawati Kandela[1], James Chou[2], Kartoa Chow[3], Reproducibility Project: Cancer Biology*

[1]Developmental Therapeutics Core, Northwestern University, Evanston, Illinois, United States; [2]LifeTein, South Plainsfield, New Jersey, United States; [3]Genentech, South San Francisco, California, United States

*For correspondence: fraser@scienceexchange.com

Group author details
Reproducibility Project: Cancer Biology
See page 13

Competing interests:
See page 13

**Abstract** The Reproducibility Project: Cancer Biology seeks to address growing concerns about reproducibility in scientific research by conducting replications of 50 papers in the field of cancer biology published between 2010 and 2012. This Registered report describes the proposed replication plan of key experiments from 'Coadministration of a tumor-penetrating peptide enhances the efficacy of cancer drugs' by Sugahara and colleagues, published in *Science* in 2010 (*Sugahara et al., 2010*). The key experiments being replicated include Figure 2 and Supplemental Figure 9A. In Figure 2, Sugahara and colleagues presented data on the tumor penetrance of doxorubicin (DOX) when co-administered with the peptide iRGD, as well as the effect of co-treatment of DOX and iRGD on tumor weight and cell death. In Supplemental Figure 9A, they tracked body weight of mice treated with DOX and iRGD to provide evidence that iRGD does not increase known DOX toxicity. The Reproducibility Project: Cancer Biology is a collaboration between the Center for Open Science and Science Exchange, and the results of the replications will be published by *eLife*.

## Introduction

$\alpha_V\beta_3$ integrin is a marker of tumor blood vessels and is targeted by the family of RGD peptides that mimic its natural ligand. This family of small peptides has been widely shown to promote targeting of a variety of therapeutics to tumor blood vessels (for review see *Danhier et al., 2012*; *Feron, 2010*). Sugahara and colleagues previously presented data showing that a novel cyclized form of this peptide, iRGD, coupled to a motif that binds to the Neuropilin-1 receptor, helped increase tissue penetrance beyond the vasculature of anti-cancer drugs when it was directly conjugated to those chemotherapies (*Sugahara et al., 2009*; *Feron, 2010*). In their 2010 study, they showed that iRGD can increase penetrance simply through co-administration with therapies, including peptide-based therapeutics, small molecule drug compounds, and nanoparticle-based therapeutics. This replication attempt will focus on the finding that iRGD co-treatment with doxorubicin (DOX) in orthotopic xenograft prostate tumors increased drug penetrance and accumulation, increased TUNEL staining for apoptotic cells, and decreased tumor volume and weight.

In Figure 2B, Sugahara and colleagues showed increased tissue permeability of DOX when co-injected with iRGD by demonstrating increased accumulation of DOX in the prostate when intravenously co-injected with iRGD. They documented little change in DOX penetrance in non-prostate tissues of the mouse. This figure is evidence of the first key point for replication: that iRGD increases tissue penetrance of the co-injected drug. This experiment will be replicated in Protocol 3.

In Figure 2C, Sugahara and colleagues showed how treatment with varying doses of DOX, with or without iRGD, affected tumor weight. They showed that treatment with 1 mg/kg DOX with iRGD caused a decrease in tumor weight similar to treatment with 3 mg/kg DOX alone. Addition of iRGD to 3 mg/kg DOX further decreased tumor weight. In Supplemental Figure 9A, they also showed that the overall body weight of mice treated with DOX and iRGD did not change as compared to mice treated with DOX alone, indicating that DOX-related weight gain was not exacerbated by the addition of iRGD. These experiments will be replicated in Protocol 4.

In Figure 2D, Sugahara and colleagues showed that treatment with DOX in combination with iRGD increased the number of TUNEL positive (i.e., apoptotic) tumor cells as compared to DOX alone. They also showed that iRGD did not increase the number of TUNEL positive cells in the heart, indicating that DOX-related cardiotoxicity is not exacerbated by co-treatment with iRGD. This experiment will be replicated in Protocol 5.

To date, the closest direct replication of these experiments has been performed by Akashi and colleagues, who tested the effects of co-administration of iRGD with the drug gemcitabine to target xenograft models of pancreatic cancer. Relative tumor volume in two pancreatic cancer cell line-derived xenografts treated with gemcitabine in combination with iRGD was significantly reduced when compared to tumor volume from xenografts treated with gemcitabine alone. Additionally, Akashi and colleagues assessed compound penetration by co-administering iRDG and the dye Evans Blue. They showed that Evans Blue dye penetrated further into tumor tissue when co-administered with iRGD than alone (*Akashi et al., 2014*). Additionally, Gu and colleagues showed that co-administration of iRGD with encapsulated paclitaxel increased nanoparticle extravasation across the blood–brain barrier, and co-administration of paclitaxel nanoparticles and iRGD led to an increase in mean survival of mice carrying intracranial gliomas (*Gu et al., 2013*). Furthermore, Pang and colleagues extended the work done in Sugahara and colleagues' 2010 paper by showing that binding of iRDG through an additional cysteine residue to plasma albumin prolonged the half-life of iRGD and increased tumor penetrance (*Pang et al., 2014*). In 2011, Ruoslahti and colleagues extended their findings to show that iRDG was also effective in increasing tumor penetrance of a novel therapeutic consisting of a nanoparticle-encapsulated peptide that homed to tumor vasculature and disrupted mitochondrial membranes, causing cell death; this co-treatment increased the survival of mice bearing glioblastomas as compared to mice treated with the nanoparticle alone (*Agemy et al., 2011*).

## Materials and methods

Unless otherwise noted, all protocol information was derived from the original paper, references from the original paper, or information obtained directly from the authors. An asterisk (*) indicates data or information provided by the Reproducibility Project: Cancer Biology core team. A hashtag (#) indicates information provided by the replicating lab.

### Protocol 1: synthesis of iRDG

This summary describes the synthesis of the peptide iRDG based on information from Sugahara and colleagues (*Sugahara et al., 2009*, *2010*). The iRGD peptide (H-Cys-Arg-Gly-Asp-Lys-Gly-Pro-Asp-Cys-NH2) will be chemically synthesized using Fmoc (9-fluorenylmethoxy carbonyl) chemistry sequence. The fully synthesized crude peptide will be cleaved from the resin and cleaned with Trifluoroacetic acid (TFA). The crude peptides will be then diethyl ether precipitated, drained, and washed. The peptides will then be amide blocked on the C terminus and cyclized by a disulfide bridge between C1 and C9. The peptides will be isolated and purified by high-performance liquid chromatography (HPLC). Fractions of greater than 95% purity will be used for the investigation. The purity and molecular weight of the peptide will be confirmed by matrix-assisted laser desorption ionization (MALDI)-time of flight mass spectrometry. Exact synthesis specifications were not originally specified; the lab will follow standard procedures for synthesis.

All data obtained from the experiment—synthesis specifications, materials, raw data, data analysis, control data, and quality control data including the MALDI-TOF mass spectrometry data—will be made publicly available, either in the published manuscript or as an open access data set available on the Open Science Framework (https://osf.io/xu1g2/).

## Protocol 2: generation of mice bearing orthotopic prostate tumors

This protocol describes how to generate mice that carry orthotopic 22Rv1 human prostate tumor xenografts.

## Sampling

- Number of mice required:
  1. For Protocol 3: 10 mice.
  2. For Protocols 4 and 5: 21 mice.
  3. Total: 31 mice.

## Materials and reagents

| Reagent | Type | Manufacturer | Catalog # | Comments |
|---|---|---|---|---|
| 22Rv1 human prostate cells | Cells | ATCC | CRL-2505 | |
| DMEM—high glucose | Medium | Sigma–Aldrich | D6429 | Original unspecified |
| FBS | Reagent | Invitrogen | 16,000 | Original unspecified |
| Penicillin/Streptomycin | Reagent | Sigma–Aldrich | P4333 | Original unspecified |
| Male 4- to 6-week-old athymic BALB/c nude mice | Mice | Harlan laboratories | Order code 069(nu)/070(nu/+) | |
| 4-0 coated vicryl suture | Materials | Ethicon | J835G | Original unspecified |
| Poly(vinylpyrrolidone)–Iodine complex | Reagent | Sigma–Aldrich | PVP-1 | Original unspecified |
| buprenonorphine | Reagent | Left to the discretion of the replicating lab and recorded later | | Original unspecified |
| isoflurane | Anesthetic | | | Original unspecified |

## Procedure

Note:

- 22Rv1 prostate cells are maintained in DMEM with 10% fetal bovine serum and penicillin/streptomycin at 37˚C/5% $CO_2$.
  1. Resuspend $1 \times 10^6$ 22Rv1 cells per 10 µl for each injection.
  2. Inject cells orthotopically into the prostate glands of male 4- to 6-week-old athymic BALB/c nude mice[#].
     a. Anesthetize mice using 2% isoflurane.
     b. Prepare skin for incision with povidone-iodine.
        i. Use sterile drapes, gloves, and instruments.
     c. Make a 5 mm midline incision over the bladder.
     d. Grasp the bladder with blunt forceps and move aside to expose ventral prostate gland.
     e. Inject cells (10 µl) into ventral prostate gland.
     f. Close incision with sutures.
     g. Inject mice subcutaneously with 0.1 mg/kg buprenonorphine immediately post operatively.
     h. Observe mice until they are awake, ambulatory, and drinking.
     i. Check mice again at the end of the day (around 5 pm).
     j. The next morning, inject mice subcutaneously with 0.1 mg/kg buprenonorphine.
     k. Remove sutures on Day 7.

## Deliverables

- Data to be collected:
  1. Mouse health records (age of mice at time of injection).
- Samples delivered for further analysis:
  1. Orthotopic tumor bearing mice for use in Protocols 2–4.

## Confirmatory analysis plan

- Statistical analysis of the replication data:
  1. None applicable.

- Meta-analysis of original and replication attempt effect sizes:
  1. Not applicable.

## Known differences from original study:

- The xenograft injection protocol is from the replicating lab. The original protocol was not specified.

## Provisions for quality control

The cell lines used in this experiment will undergo STR profiling to confirm their identity and will be sent for mycoplasma testing to ensure there is no contamination. Additionally, cells used for xenograft injection will be screened against a Rodent Pathogen Panel to ensure no contamination prior to injection. All data obtained from the experiment—raw data, data analysis, control data, and quality control data—will be made publicly available, either in the published manuscript or as an open access data set available on the Open Science Framework (https://osf.io/xu1g2/).

- A lab with experience in prostate gland tumor xenografts will perform the experiment.

## Protocol 3: quantifying the amount of Dox present in tumor tissue and major organs in mice treated with Dox with or without iRGD

This protocol describes how to treat mice bearing human 22Rv1 prostate tumors from Protocol 2 with DOX and/or iRGD, harvest the tumors and assess DOX penetrance by measuring absorbance at 490 nm ($OD_{490}$), as seen in Figure 2B.

### Sampling

- This experiment will analyze at least 3 mice per group for a final power of 97.2%.
  1. See power calculations section for details.
- The experiment consists of two cohorts:
  1. Cohort 1: mice treated with DOX and PBS.
    A. N = 4.
      - To buffer against unexpected mouse deaths, 4 mice bearing tumors will be treated.
  2. Cohort 2: mice treated with DOX and iRGD.
    A. N = 4.
      - To buffer against unexpected mouse deaths, 4 mice bearing tumors will be treated.
  3. Cohort 3: untreated mice.
    A. N = 2.

### Materials and reagents

| Reagent | Type | Manufacturer | Catalog # | Comments |
|---|---|---|---|---|
| Orthotopic tumor bearing mice | Mice | From Protocol 2 | | |
| Doxorubicin hydrochloride | Drug | Sigma–Aldrich | D1515 | Original unspecified |
| iRGD | Peptide | From Protocol 1 | | |
| 30 G needle/syringe for IV injection of drugs (step 3) | Materials | BD Biosciences | 305106 | |
| Bovine Serum Albumin (BSA) | Reagent | Sigma–Aldrich | A2153 | Original unspecified |
| Chloroform | Reagent | Sigma–Aldrich | C2432 | Original unspecified |
| Isopropyl alcohol | Reagent | Left to the discretion of the replicating lab and recorded later | | Original unspecified |
| 15-ml conical tubes | Materials | Corning Life Sciences | 352095 | Original unspecified |
| Dulbecco's Phosphate Buffered Saline (PBS) | Reagent | Sigma–Aldrich | D8537 | Original unspecified |
| Animal feeding needle, 24 G, L × diam. 1 in. × 1.25 mm, ball | Material | Sigma–Aldrich | CAD7900 | Original unspecified |
| 18 G syringe needles | Material | Sigma–Aldrich | Z192554 | Original unspecified |
| Sodium dodecyl sulfate solution (10%) | Reagent | Sigma–Aldrich | 71,736 | Original unspecified |
| Sulfuric acid (H2S04) | Reagent | Sigma–Aldrich | 339741 | Original unspecified |
| 8453 UV-Vis spectrophotometer | Instrument | Agilent | | Original unspecified |
| DMEM—high glucose | Medium | Sigma–Aldrich | D6429 | Original unspecified |

## Procedure

1. Generate tumor bearing mice as per Protocol 2.
2. Allow tumors to grow for 2 weeks from time of injection.
3. Inject mice with drugs in combination:
   a. On day of injection, randomly assign the 10 mice into the two treatment groups and the untreated group.
      i. Assign each mouse a number 1 through 10.
      ii. After mice have been assigned numbers, enter the treatment labels (4 labels as Negative control, 4 labels as Experimental, and 2 labels as Untreated), and randomize 10 subjects into 1 block using www.randomization.com. Record seed number.
   b. Negative control: inject mice intravenously with 10 mg/kg Dox suspended in 100 µl PBS.
   c. Experimental: inject mice intravenously with 10 mg/kg Dox and 4 µmol/kg iRGD suspended in 100 µl PBS.
   d. Untreated mice receive no injections.
4. 1 hour later, sacrifice mice and excise tissues:
   a. Deeply anesthetize the mice with isoflurane.
   b. Perfuse through the heart with PBS + 1% BSA#.
      i. Place the deeply anesthetized mouse in a heated cage for 10 min.
      ii. Secure the mouse in the supine position by taping the paws to a Styrofoam work surface.
      iii. Make an incision through the skin with surgical scissors along the thoracic midline from just beneath the xiphoid process to the clavicle. Make two additional skin incisions from the xiphoid process along the base of the ventral ribcage laterally.
      iv. Reflect the two flaps of skin rostrally and laterally making sure to expose the thoracic field completely.
      v. Grasp the cartilage of the xiphoid process with blunt forceps and raise it slightly to insert pointed scissors. Cut through the thoracic musculature and ribcage between the breastbone and medial rib insertion points and extend the incision rostrally to the level of the clavicles.
      vi. Separate the diaphragm from the chest wall on both sides with scissors.
      vii. Pin with 18 G needles the reflected ribcage laterally to expose the heart.
      viii. Gently grasp the pericardial sac with blunt forceps and tear it fully.
      ix. Secure the beating heart with blunt forceps and make a 1–2 mm incision in the left ventricle. Immediately insert a 24 G × 25.4 mm animal feeding needle. The tip is bulbous and will not damage the heart. Thread the feeding needle into the base of the aortic arch using a dissecting microscope. Clamp the needle base to the left ventricle above the incision site using a hemostat.
      x. Cut the right atrium with scissors and at the first sign of blood flow, begin infusion of DMEM containing 1% BSA (stage 1 perfusate).
         1. Use gravity driven perfusion at a rate of 3 mls per minute.
      xi. Continue perfusing the body until the fluid exiting the right atrium is entirely clear.
      xii. Ensure that organs of interest become pale; if an organ does not become pale, exclude the organ from further analysis.
   c. Excise prostate tumor tissue, liver, spleen, pancreas, heart, lung, kidneys, and brain.
5. Homogenize each tissue separately in 1% sodium dodecyl sulfate and 1 mM $H_2SO_4$ in water.
   a. Homogenize 100-mg tissue in 0.5 ml solution using a mortar and pestle#.
6. Add 2 ml of chloroform:isopropyl alcohol (1:1, vol/vol).
7. Vortex in 15-ml conical tubes and run through freeze/thaw cycles.
   a. #Place tubes on dry ice for 1 min to freeze, then thaw for 5 min in a 25°C waterbath.
8. Centrifuge samples at 14,000×g for 15 min at #room temperature.
   a. Store samples at 4°C until ready for Step 9.
   b. For the two untreated mice, combine their samples to create the blank reference for each tissue; that is, combine tumor with tumor, liver with liver, etc.
9. Measure the $OD_{490}$ of the organic phase (the lowest phase).
   a. For each measurement, blank with appropriate tissue homogenate (tumor, liver, spleen, etc) from the untreated control mice.
   b. Calculate the fold change in Dox level from mice treated with iRGD by dividing by the absorbance reading of mice treated with Dox alone.
   c. Graph the fold change by tissue.

## Deliverables

- Data to be collected:

1. Raw readings of $OD_{490}$ absorbance of each sample.
2. Graph of DOX accumulation with iRGD or PBS per organ (compare to Figure 2B).

## Confirmatory analysis plan

- Statistical analysis of the replication data:
  1. At the time of analysis, we will perform the Shapiro–Wilk test and generate a quantile–quantile (q–q) plot to attempt to assess the normality of the data and also perform Levene's test to assess homoscedasiticity. If the data appear skewed, we will attempt a transformation in order to proceed with the proposed statistical analysis listed below and possibly perform the appropriate non-parametric test.
     A. Compare the level of Dox + iRGD to level of Dox alone in tumor tissue.
        - Unpaired two-tailed Student's *t*-test.
          1. Original analysis.
- Meta-analysis of original and replication attempt effect sizes:
  1. This replication attempt will perform the statistical analysis listed above, compute the effects sizes, compare them against the reported effect size in the original paper, and use a meta-analytic approach to combine the original and replication effects, which will be presented as a forest plot.

## Known differences from original study

- Details noted with a hashtag (#) were provided by the replicating lab.

## Provisions for quality control

Tissue homogenate from untreated mice will be used to blank the spectrophotometer. Mice will be randomly assigned to treatment groups. All data obtained from the experiment—raw data, data analysis, control data and quality control data—will be made publicly available, either in the published manuscript or as an open access data set available on the Open Science Framework (https://osf.io/xu1g2/).

- A lab with experience in prostate gland tumor xenografts will perform the experiment.

## Protocol 4: effect of Dox alone or Dox in combination with iRGD on tumor growth and total body weight

This protocol describes how to treat mice bearing human 22Rv1 prostate tumors from Protocol 2 with DOX and/or iRGD, monitor body weight and then assess tumor weight, as seen in Figure 2C and Supplemental Figure 9A.

### Sampling

- This experiment will analyze at least 6 mice per group for a final power of 93.5%.
  1. See power calculations section for details.
- The experiment consists of three cohorts:
  1. Cohort 1: mice treated with PBS alone.
     A. N = 7.
        - To buffer against unexpected mouse deaths, 7 mice bearing tumors will be treated.
  2. Cohort 2: mice treated with 1 mg/kg Dox and PBS.
     A. N = 7.
        - To buffer against unexpected mouse deaths, 7 mice bearing tumors will be treated.
  3. Cohort 3: mice treated with 1 mg/kg Dox and 4 μmol/kg iRGD.
     A. N = 7.
        - To buffer against unexpected mouse deaths, 7 mice bearing tumors will be treated.

### Materials and reagents

| Reagent | Type | Manufacturer | Catalog # | Comments |
|---|---|---|---|---|
| Orthotopic tumor bearing mice | Mice | From Protocol 2 | | |
| Doxorubicin hydrochloride | Drug | Sigma–Aldrich | D1515 | Vehicle is PBS |
| iRGD | Peptide | From Protocol 1 | | Vehicle is PBS |
| Paraformaldehyde | Reagent | Sigma–Aldrich | 158127 | Original unspecified |
| Paraffin | Reagent | Specific brand left to the discretion of the replicating lab and will be recorded later | | Original unspecified |
| Bovine Serum Albumin (BSA) | Reagent | Sigma–Aldrich | A2153 | Original unspecified |
| Dulbecco's Phosphate Buffered Saline (PBS) | Reagent | Sigma–Aldrich | D8537 | Original unspecified |

## Procedure

1. Generate tumor bearing mice as per Protocol 2.
2. Allow tumors to grow for 2 weeks from time of injection.
3. Inject mice with drugs according to their cohort; this is Day 0.
   a. On day of injection, randomly assign the 21 mice into the three treatment groups.
      i. Assign each mouse a number 1 through 21.
      ii. After mice have been assigned numbers, enter the treatment labels (Cohort 1, Cohort 2, and Cohort 3) and randomize 3 subjects into 7 blocks using www.randomization.com. Record seed number.
   b. Cohort 1: mice treated with PBS alone.
   c. Cohort 2: mice treated with 1 mg/kg Dox and PBS.
   d. Cohort 3: mice treated with 1 mg/kg Dox and 4 µmol/kg iRGD.
4. Repeat injection every other day for 24 days.
5. Weigh mice every 4 days, starting on Day 0.
6. After 24 days of treatment, harvest tissue.
   a. Perfuse mice as outlined in Protocol 3 Steps 4a–b.
   b. Excise prostate tumor tissue and heart tissue.
7. Weigh tumor tissue.
8. Process, embed, and section tissue.
   a. Fix tumor and heart tissue in 4% paraformaldehyde overnight at 4°C.
   b. Cut each tumor and each heart in half.
   c. #Dehydrate tissue and infiltrate with paraffin.
   d. #Embed in paraffin.
      i. Use one half of each tumor or heart to perform the sectioning below. Hold the other half in reserve in case more sections are needed later.
   e. Cut at least 7 5-µm thick sections spaced throughout the tumor or heart halves and mount on glass slides (i.e., the sections should not be serial).
      i. Tumor and heart sections to be used in Protocol 5.

## Deliverables

- Data to be collected:
  1. Record of the drug treatment regimen and weight of each tumor for each mouse.
  2. Raw values for mouse body weight at time points during treatment.
  3. Graph of tumor weight by drug treatment in grams (compare to Figure 2C).
  4. Graph of change in body weight as a percentage of body weight on day 0 (compare to Supplemental Figure 9A).
- Samples delivered for further analysis.
  1. Tumor and heart tissue processed to sections for further analysis (see Protocol 5).

## Confirmatory analysis plan

- Statistical analysis of the replication data:
  1. At the time of analysis, we will perform the Shapiro–Wilk test and generate a quantile–quantile (q–q) plot to attempt to assess the normality of the data and also perform Levene's test to assess homoscedasiticity. If the data appear skewed, we will attempt a transformation in order to proceed with the proposed statistical analysis listed below and possibly perform the appropriate non-parametric test.
   A. Tumor weights in each cohort (as seen in Figure 2C).
      - One-way ANOVA followed by Fisher's LSD $t$-tests for the following comparisons:
        1. 1 mg/kg Dox vs 1 mg/kg Dox and 4 µmol/kg iRGD.
   A. Body weight shift (as seen in Supplemental Figure 9A).
      - One-way ANOVA on Day 24 time points.
        1. As seen in the original analysis.
      - Additional analysis: one-way ANOVA of calculated area under the curve of mouse body weight from each cohort followed by Fisher's LSD corrected $t$-tests for the following comparison:
        1. 1 mg/kg Dox vs 1 mg/kg Dox and 4 µmol/kg iRGD.
- Meta-analysis of original and replication attempt effect sizes:
  1. This replication attempt will perform the statistical analysis listed above, compute the effects sizes, compare them against the reported effect size in the original paper and use

a meta-analytic approach to combine the original and replication effects, which will be presented as a forest plot.

## Known differences from original study

- The replication study will be restricted to examining the following groups:
  1. No dox/no peptide.
  2. 1 mg/kg Dox.
  3. 1 mg/kg Dox/iRGD.
- The tumor tissue will be embedded in paraffin for paraffin sectioning rather than in OCT for cryosectioning.

## Provisions for quality control

Mice will be randomly assigned to treatment groups. All data obtained from the experiment—raw data, data analysis, control data, and quality control data—will be made publicly available, either in the published manuscript or as an open access data set available on the Open Science Framework (https://osf.io/xu1g2/).

- A lab with experience in prostate gland tumor xenografts will perform the experiment.

## Protocol 5: assessment of TUNEL staining of tumor and heart tissue after drug treatment

This protocol describes how to assess cell death via TUNEL staining in prostate tumors derived from 22Rv1 xenografts treated with DOX and/or iRGD, as seen in Figure 2D.

### Sampling

- This protocol uses tissues derived from Protocol 4.
  1. This experiment will analyze 6 tumors per group, for a final power of 88.8%.

### Materials and reagents

| Reagent | Type | Manufacturer | Catalog # | Comments |
|---|---|---|---|---|
| Sections of tumor and heart tissue | Tissue | From Protocol 4 | | |
| In Situ Cell Death Detection Kit POD | Reagent | Roche Applied Science | 11684817910 | |
| Microscope | Instrument | Olympus | BX40 | |
| Scanscope scanner | Instrument | Aperio | CM-1 | |
| ImageJ | Software | NIH | | |

### Procedure

Note: This protocol uses tumor and heart tissues derived from Protocol 4.

1. Perform TUNEL staining of tumor and heart tissue sections with the In Situ Cell Death Detection Kit POD according to the manufacturer's instructions.
   a. Include manufacturer's recommended controls:
      i. Positive control: incubate with label solution instead of with TUNEL reaction mixture.
      ii. Negative control: incubate with micrococcal nuclease prior to labeling procedure.
   b. Stain a total of 7 slides for each tissue; 5 for analysis, one negative control, one positive control.
2. Scan the stained sections with a Scanscope CM-1 scanner and quantify areas of TUNEL positive staining with ImageJ software.
   a. *Image 5 random fields at 40× per section and image 5 sections per tumor and per heart.
   b. If sections are unable to be imaged due to autofluorescence or damage during the staining procedure, take images, and exclude from analysis with indicated reason.

## Deliverables

- Data to be collected:
  1. All images taken for all tumors and treatment groups, including control images.
  2. Raw numbers from ImageJ analysis for each field and section of each tumor and heart.
  3. Determine the ratio of TUNEL staining as fold change relative to staining in the tumors treated with PBS only. Graph by treatment regimen.

## Confirmatory analysis plan

- Statistical analysis of the replication data:
  1. At the time of analysis, we will perform the Shapiro–Wilk test and generate a quantile–quantile (q–q) plot to attempt to assess the normality of the data and also perform Levene's test to assess homoscedasiticity. If the data appear skewed, we will attempt a transformation in order to proceed with the proposed statistical analysis listed below and possibly perform the appropriate non-parametric test.

     A. Compare TUNEL positive area in tumor across all conditions.
       - Bonferroni corrected one-way ANOVA followed by Bonferroni corrected $t$-tests for the following comparisons:
         1. Mice treated with 1 mg/kg Dox and PBS vs mice treated with 1 mg/kg Dox and 4 μmol/kg iRGD.

     A. Compare TUNEL positive area in heart across all conditions.
       - Bonferroni corrected one-way ANOVA followed by Bonferroni corrected $t$-tests for the following comparisons:
         1. Mice treated with 1 mg/kg Dox and PBS vs mice treated with 1 mg/kg Dox and 4 μmol/kg iRGD.

- Meta-analysis of original and replication attempt effect sizes:
  1. This replication attempt will perform the statistical analysis listed above, compute the effects sizes, compare them against the reported effect size in the original paper, and use a meta-analytic approach to combine the original and replication effects, which will be presented as a forest plot.

## Known differences from original study

- The replication study will be restricted to examining the following groups:
  1. No dox/no peptide.
  2. 1 mg/kg Dox/no peptide.
  3. 1 mg/kg Dox/iRGD.

## Provisions for quality control

Manufacturer recommended positive and negative controls will be used when performing TUNEL staining of tumor and heart tissues. All data obtained from the experiment—raw data, data analysis, control data, and quality control data—will be made publicly available, either in the published manuscript or as an open access data set available on the Open Science Framework (https://osf.io/xu1g2/).

- A lab with experience in prostate gland tumor xenografts will perform the experiment.

## Power calculations

## Protocol 1

- Not applicable.

## Protocol 2

- Not applicable.

## Protocol 3

### Summary of original data

- Note: values estimated from original graph.

| Figure 2B: DOX accumulation fold free dox | | Mean | SEM | SD | N |
|---|---|---|---|---|---|
| Tumor | DOX | 1 | 0.48 | 0.83 | 3 |
| | DOX + iRGD | 7.15 | 1.05 | 1.82 | 3 |
| Liver | DOX | 1 | 0.6 | 1.04 | 3 |
| | DOX + iRGD | 1.51 | 0.67 | 1.16 | 3 |
| Spleen | DOX | 1 | 0.4 | 0.69 | 3 |
| | DOX + iRGD | 0.36 | 0.72 | 1.25 | 3 |
| Pancreas | DOX | 1 | 0.69 | 1.20 | 3 |
| | DOX + iRGD | 0.15 | 0.09 | 0.16 | 3 |
| Heart | DOX | 1 | 0.57 | 0.99 | 3 |
| | DOX + iRGD | 0.21 | 0.29 | 0.50 | 3 |
| Lung | DOX | 1 | 0.67 | 1.16 | 3 |
| | DOX + iRGD | 0.77 | 0.66 | 1.14 | 3 |
| Kidney | DOX | 1 | 0.12 | 0.21 | 3 |
| | DOX + iRGD | 1 | 0.02 | 0.03 | 3 |
| Brain | DOX | 1 | 0.19 | 0.33 | 3 |
| | DOX + iRGD | 0.44 | 0.23 | 0.40 | 3 |

Stdev was calculated using formula SD = SEM*(SQRT n).

### Test family

- Unpaired two-tailed Student's $t$-test comparing tumor free Dox to tumor Dox + iRGD.
  1. As seen in the original analysis.

### Power calculations

- Power calculations performed using G*power software (*Faul et al., 2007*).
- $\alpha = 0.05$.

| Group 1 vs | Group 2 | Effect size | A priori power | Group 1 sample size | Group 2 sample size |
|---|---|---|---|---|---|
| Dox alone in tumor tissue | Dox + iRGD in tumor tissue | 4.348000 | 97.2% | 3 | 3 |

## Protocol 4

### Summary of original data

- Note: values estimated from original graph.

| Figure 2C: Tumor weight | | Mean | SEM | SD | N |
|---|---|---|---|---|---|
| Free Dox | Peptide | | | | |
| None | None | 1.19 | 0.07 | 0.22 | 10 |
| 1 mg/kg | None | 0.817 | 0.093 | 0.29 | 10 |
| 1 mg/kg | iRGD | 0.35 | 0.02 | 0.06 | 10 |

Stdev was calculated using formula SD = SEM*(SQRT n).

## Test family

- One-way ANOVA followed by Fisher's LSD *t*-tests for the following comparison:
  1. Mice treated with 1 mg/kg Dox and PBS vs mice treated with 1 mg/kg Dox and iRGD.

## Power calculations

- F statistic and partial η2 performed with R software (3.1.2) (*R Core team, 2014*).
- Power calculations performed using G*power software (*Faul et al., 2007*).
- α = 0.05.

**One-way ANOVA**

| F (2, 27) | p-value | Partial η2 | Effect size f | A priori power | Sample size per group |
|---|---|---|---|---|---|
| 39.045 | <0.0001 | 0.74308 | 1.700665 | 94.1%* | 3* |

*Due to power calculations for Figure 2D, we will be using 6 tumors per group, for an achieved power of 99.9%.

**T-test**

| Group 1 vs | Group 2 | Effect size d | A priori power | Group 1 sample size | Group 2 sample size |
|---|---|---|---|---|---|
| 1 mg/kg Dox and PBS | 1 mg/kg Dox and iRGD | 2.230140 | 86.9%* | 5* | 5* |

*Due to power calculations for Figure 2D, we will be using 6 tumors per group, for an achieved power of 93.5%.

## Summary of original data

- Note: values estimated from original graph

| Supp. Figure 9A | | Mean body weight shift (%) | SEM | SD | N |
|---|---|---|---|---|---|
| PBS | Day 0 | 100 | 0 | 0 | 10 |
| | Day 4 | 100.639 | 1.532 | 4.844609375 | 10 |
| | Day 8 | 100.958 | 1.564 | 4.945802261 | 10 |
| | Day 12 | 102.298 | 2.489 | 7.870909096 | 10 |
| | Day 16 | 104.5 | 1.532 | 4.844609375 | 10 |
| | Day 20 | 105.585 | 2.202 | 6.963335408 | 10 |
| | Day 24 | 105.904 | 1.564 | 4.945802261 | 10 |
| 1 mg/kg Dox | Day 0 | 100 | 0 | 0 | 10 |
| | Day 4 | 101.213 | 0.957 | 3.026299721 | 10 |
| | Day 8 | 103.287 | 1.485 | 4.695982325 | 10 |
| | Day 12 | 103.335 | 1.484 | 4.692820048 | 10 |
| | Day 16 | 105.394 | 2.059 | 6.511129702 | 10 |
| | Day 20 | 105.585 | 2.202 | 6.963335408 | 10 |
| | Day 24 | 106.734 | 2.346 | 7.418703391 | 10 |
| 1 mg/kg dox + iRGD | Day 0 | 100 | 0 | 0 | 10 |
| | Day 4 | 99.203 | 2.202 | 6.963335408 | 10 |
| | Day 8 | 99.968 | 1.181 | 3.734649917 | 10 |
| | Day 12 | 99.84 | 1.309 | 4.139421457 | 10 |
| | Day 16 | 101.692 | 1.438 | 4.547355275 | 10 |
| | Day 20 | 102.553 | 1.564 | 4.945802261 | 10 |
| | Day 24 | 103.32 | 1.628 | 5.148188031 | 10 |

## Test family

- One-way ANOVA on Day 24 time points.
  1. As seen in the original analysis.
- Additional analysis: one-way ANOVA of calculated area under the curve of mouse body weight from each cohort followed by Fisher's LSD corrected t-tests for the following comparison:
  1. 1 mg/kg Dox vs 1 mg/kg Dox and 4 μmol/kg iRGD.

## Sensitivity calculations

- F statistic and partial η2 performed with R software (3.1.2) (*R Core team, 2014*).
- Sensitivity calculations performed using G*power software (*Faul et al., 2007*).
- α = 0.05.
- Sample size derived from Figure 2D (Protocol 5) power calculations.
- One way ANOVA on day 24 time points.

**One-way ANOVA**

| F (2, 27) | Partial η2 | Reported effect size f | Detectable effect size f | A priori power | Sample size per group |
|-----------|-----------|------------------------|--------------------------|----------------|------------------------|
| 0.0487 | 0.003597 | 0.060083 | 0.811282 | 80.0% | 6 |

- Additional analysis: area under the curve performed with R software (3.1.2) (*R Core team, 2014*).

**Area under the curve**

| | Mean | SD | N |
|---|------|-----|---|
| PBS | 2467.728 | 127.7687 | 10 |
| 1 mg/kg dox | 2488.724 | 118.3957 | 10 |
| 1 mg/kg dox + iRGD | 2419.664 | 107.6186 | 10 |

**One-way ANOVA**

| F (2, 27) | Partial η2 | Reported effect size f | Detectable effect size f | A priori power | Sample size per group |
|-----------|-----------|------------------------|--------------------------|----------------|------------------------|
| 0.8969 | 0.062297 | 0.257751 | 0.811282 | 80.0% | 6 |

**Fisher's LSD corrected t-test**

| Group 1 | Group 2 | Reported effect size d | Detectable effect size d | A priori power | Sample size per group |
|---------|---------|------------------------|--------------------------|----------------|------------------------|
| 1 mg/kg dox | 1 mg/kg dox + iRGD | 0.610418 | 1.795541 | 80.0% | 6 |

## Protocol 5
## Summary of original data

- Note: Values estimated from original graph.

| Figure 2D: TUNEL staining of tumor and heart | | | Mean | SEM | SD | N |
|---|---|---|------|-----|-----|---|
| **Free Dox** | **Peptide** | **Tissue** | | | | |
| None | None | Tumor | 1 | 0.1 | 0.32 | 10 |
| | | Heart | 1 | 0.27 | 0.85 | 10 |
| 1 mg/kg | None | Tumor | 1.4 | 0.17 | 0.54 | 10 |
| | | Heart | 1.65 | 0.28 | 0.89 | 10 |
| 1 mg/kg | iRGD | Tumor | 2.58 | 0.2 | 0.63 | 10 |
| | | Heart | 1.7 | 0.31 | 0.98 | 10 |

- Stdev was calculated using formula SD = SEM*(SQRT n).

## Test family

- Bonferroni corrected one-way ANOVAs followed by Bonferroni's corrected *t*-tests for the following comparison:
  1. Tumor tissue from mice treated with 1 mg/kg Dox and PBS vs mice treated with 1 mg/kg Dox and iRGD.

## Power calculations

- F statistic and partial η2 performed with R software (3.1.2) (*R Core Team, 2014*).
- Power calculations performed using G*power software (*Faul et al., 2007*).
- α = 0.025.
- Sample size calculations performed for tumor tissue and sensitivity calculations for heart tissue.

**One-way ANOVA**

| F (2, 27) | Partial η2 | Reported effect size f | Detectable effect size f | A priori power | Sample size per group |
|---|---|---|---|---|---|
| **Tumor** | | | | | |
| 25.596 | 0.654698 | 1.376959 | n/a | 89.5%* | 4* |
| **Heart** | | | | | |
| 1.8485 | 0.120434 | 0.370033 | 0.908639 | 80.0% | 6 |

*With a sample size of 6 per group, achieved power is 99.4%.

**Bonferroni corrected t-tests**

| Group 1 vs | Group 2 | Reported effect size d | Detectable effect size d | A priori power | Group 1 sample size | Group 2 sample size |
|---|---|---|---|---|---|---|
| **Tumor** | | | | | | |
| 1 mg/kg Dox alone | 1 mg/kg Dox + iRGD | 2.298170 | n/a | 88.8% | 6 | 6 |
| **Heart** | | | | | | |
| 1 mg/kg Dox alone | 1 mg/kg Dox + iRGD | 0.055048 | 2.044185 | 80.0% | 6 | 6 |

# Acknowledgements

The Reproducibility Project: Cancer Biology core team would like to thank the following companies for generously donating reagents to the Reproducibility Project: Cancer Biology; American Type Culture Collection (ATCC), BioLegend, Cell Signaling Technology, Charles River Laboratories, Corning Incorporated, DDC Medical, EMD Millipore, Harlan Laboratories, LI-COR Biosciences, Mirus Bio, Novus Biologicals, Sigma–Aldrich, and System Biosciences (SBI).

# Additional information

### Group author details

**Reproducibility Project: Cancer Biology**

Elizabeth Iorns: Science Exchange, Palo Alto, California; William Gunn: Mendeley, London, United Kingdom; Fraser Tan: Science Exchange, Palo Alto, California; Joelle Lomax: Science Exchange, Palo Alto, California; Nicole Perfito: Science Exchange, Palo Alto, California; Timothy Errington: Center for Open Science, Charlottesville, Virginia

### Competing interests

RP:CB: We disclose that EI, FT, JL, and NP are employed by and hold shares in Science Exchange Inc. The experiments presented in this manuscript will be conducted by IK at the Developmental

Therapeutics Core and JC at Lifetein, which are Science Exchange labs. The other authors declare that no competing interests exist.

## Funding

| Funder | Author |
|---|---|
| Laura and John Arnold foundation | Reproducibility Project: Cancer Biology |

The Reproducibility Project: Cancer Biology is funded by the Laura and John Arnold Foundation, provided to the Center for Open Science in collaboration with Science Exchange. The funder had no role in study design or the decision to submit the work for publication.

## Author contributions

IK, JC, KC, Drafting or revising the article; RP:CB, Conception and design, Drafting or revising the article

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
