## [Decision Letter]

Thank you for sending your work entitled “Registered report: Coadministration of a Tumor-Penetrating Peptide Enhances the Efficacy of Cancer Drugs” for consideration at *eLife*. Your article has been favorably evaluated by Stylianos Antonarakis (Senior editor), a Reviewing editor, and two reviewers, one of whom, Erkki Ruoslahti, has agreed to share his identity.

The Reviewing editor and the reviewers discussed their comments before we reached this decision, and the Reviewing editor has assembled the following comments to help you prepare a revised submission.

Overall, the reviewers have found the experimental plan and statistical analysis to be reasonable. One particular concern is the number of animals. While they are sufficient for one experiment, ideally one would like to see the experiment to be repeated at least once. The reviewers have suggested the following corrections/improvements to be considered in a revision:

*Reviewer #1*:

This is a reasonable plan, but a few issues should be dealt with:

1) It would be appropriate to cite Agemy et al. PNAS 108:17450-17455 (2011) PMCID: PMC3198371 in the Introduction, as it reports iRGD work partially conducted at the Salk Institute.

2) The peptide sequence in the subsection headed “Protocol 1: Synthesis of iRDG” is incorrect.

3) The iRGD peptide sequence in the subsection headed “Protocol 1: Synthesis of iRDG” is shown with unblocked N- and C-termini. The original paper is silent on whether they were blocked or not, but in the subject work, Sugahara et al. used iRGD that was amide-blocked at the C-terminus. Although the unblocked peptide is also known to be active, it would be safer to block the C-terminus.

4) The tumor line Sugahara et al. used is 22Rv1 (ATCC #CRL-2505). Here it is said to be RRv1 and the ATCC number given corresponds to 22Rv1.

5) Protocol 3, item 4b: In the protocol, the mice are first euthanized but later on the mice are said to have a beating heart. The perfusion should be done under deep anesthesia with beating heart, not after euthanasia. Also, gravity driven perfusion at about 3 ml/min should be used, and the solution should be DMEM containing 1% BSA (not heparinzed saline). Besides making sure that fluid exciting the right atrium becomes clear, one should see a change in blood content of organs; if an organ does not become pale, that tissue should not be used for further analyses.

6) The most serious flaw in the plan concerns the number of mice to be used. They do not allow for any repeat experiments. To rely on one experiment in a study that will presumably be highly publicized and could make or break careers does not seem appropriate.

*Reviewer #2*:

This protocol plans to replicate some of the key findings of Sugahara and colleagues regarding the benefits of DOX + iRGD in relation to prostate tumor in mice.

The power calculations are fine.

Given that the proposed work requires a small number of mice, it may be worthwhile to consider nonparametric analysis methods in addition to the proposed t test and ANOVA approaches during data analyses.

The investigators plan to use mixed effects model ANOVA as an additional exploratory analysis in protocol 3. It is not clear why this analysis is needed. It is possible that a mixed effects model would be stretching the data too far due to the small number of mice. My suggestion would be to remove this mixed effects analysis plan (unless there is a compelling reason to pursue this).

---

## [Author Response]

Reviewer #1:

*This is a reasonable plan, but a few issues should be dealt with*:

*1) It would be appropriate to cite Agemy et al. PNAS 108:17450-17455 (2011) PMCID: PMC3198371 in the Introduction, as it reports iRGD work partially conducted at the Salk Institute*.

Thank you for sharing this manuscript. We have updated the Introduction to include this reference.

*2) The peptide sequence in the subsection headed “Protocol 1: Synthesis of iRDG” is incorrect*.

Thank you for catching this error. We have amended the manuscript to correctly state the first amino acid as Gly instead of Cys.

*3) The iRGD peptide sequence in the subsection headed “Protocol 1: Synthesis of iRDG” is shown with unblocked N- and C-termini. The original paper is silent on whether they were blocked or not, but in the subject work, Sugahara et al. used iRGD that was amide-blocked at the C-terminus. Although the unblocked peptide is also known to be active, it would be safer to block the C-terminus*.

Thank you for this critical feedback. We have updated the manuscript to include this step in the synthesis of the peptide.

The lab preparing the iRGD peptide, having read the original paper and others describing iRGD, wanted to know if the iRGD used in Sugahara and colleagues’ 2010 paper was cyclized, specifically with a C1-C9 disulfide bridge. Would the editors be so kind as to pass this question back to the reviewers for their feedback?

[Editors’ note: the authors were informed that iRGD is a cyclic peptide with a disulfide bond between C1 and C9, and that the linear version of the peptide is inactive.]

*4) The tumor line Sugahara et al. used is 22Rv1 (ATCC #CRL-2505). Here it is said to be RRv1 and the ATCC number given corresponds to 22Rv1*.

We have updated the manuscript to refer to the cell line as 22Rv1.

*5) Protocol 3, item 4b: In the protocol, the mice are first euthanized but later on the mice are said to have a beating heart. The perfusion should be done under deep anesthesia with beating heart, not after euthanasia. Also, gravity driven perfusion at about 3 ml/min should be used, and the solution should be DMEM containing 1% BSA (not heparinzed saline). Besides making sure that fluid exciting the right atrium becomes clear, one should see a change in blood content of organs; if an organ does not become pale, that tissue should not be used for further analyses*.

Thank you for sharing these important details regarding the proper treatment of the mice. We have updated the manuscript to reflect these changes.

*6) The most serious flaw in the plan concerns the number of mice to be used. They do not allow for any repeat experiments. To rely on one experiment in a study that will presumably be highly publicized and could make or break careers does not seem appropriate*.

While it is accurate that the replication will not be conducted multiple times, the plan includes multiple steps to ensure a high level of quality control and transparency in the process. Part of this process is to ensure the replication experiments are designed to have a low type I and type II error rate for detecting the originally reported effect size. Additionally, any known difference from the original experiments and replication experiments are detailed and whether any of these might impact the replication. This also includes valuable insight from the authors of the originally reported results. Additionally, all data collected during the experimental procedure will be recorded and made publically available to allow the collective experimental results to be evaluated. This project aims to evaluate the predictors of reproducing a subset of the published literature. Thus, the focus is on a collection of experimental outcomes, and the factors associated with them, and not the conclusions from any given paper, which are based on multiple experiments and models. Additionally, a failure to replicate does not mean the original result was a false positive, just like an ability to replicate the original result does not mean the original interpretation is correct.

Reviewer #2:

*Given that the proposed work requires a small number of mice, it may be worthwhile to consider nonparametric analysis methods in addition to the proposed t test and ANOVA approaches during data analyses*.

We have added language to the Confirmatory Analysis plan section detailing that we will assess the normality of the data before proceeding with the proposed parametric tests, and if the data do not conform to the assumptions for the proposed tests, we will use the appropriate non-parametric tests instead.

*The investigators plan to use mixed effects model ANOVA as an additional exploratory analysis in protocol 3. It is not clear why this analysis is needed. It is possible that a mixed effects model would be stretching the data too far due to the small number of mice. My suggestion would be to remove this mixed effects analysis plan (unless there is a compelling reason to pursue this)*.

Thank you for the feedback. We agree and have removed that additional analysis from the manuscript.